# Sevoflurane Preconditioning Rescues PKMζ Gene Expression from Broad Hypoxia-Induced mRNA Downregulation Correlating with Improved Neuronal Recovery

**DOI:** 10.3390/neurosci6010009

**Published:** 2025-01-28

**Authors:** Joan Y. Hou, Kim D. Allen, A. Iván Hernandez, James E. Cottrell, Ira S. Kass

**Affiliations:** 1Anesthesiology Department, State University of New York Downstate Health Sciences University, Brooklyn, NY 11203, USA; 2Pathology Department, State University of New York Downstate Health Sciences University, Brooklyn, NY 11203, USA; kallen@mec.cuny.edu (K.D.A.); ivan.hernandez@downstate.edu (A.I.H.); 3Department of Biology, Medgar Evers College, Brooklyn, NY 11225, USA; 4Anesthesiology Department, Physiology and Pharmacology Department, State University of New York Downstate Health Sciences University, Brooklyn, NY 11203, USA

**Keywords:** anesthesia, sevoflurane, hypoxia, protein kinase mRNA, anti-apoptotic mRNA, transcriptional downregulation

## Abstract

Hypoxia due to stroke is a major cause of neuronal damage, leading to loss of cognition and other brain functions. Sevoflurane preconditioning improves recovery after hypoxia. Hypoxia interferes with protein expression at the translational level; however, its effect on mRNA levels for neuronal protein kinase and anti-apoptotic genes is unclear. To investigate the link between sevoflurane preconditioning and gene expression, hippocampal slices were treated with 4% sevoflurane for 15 min, a 5 min washout, 10 min of hypoxia, and 60 min of recovery. We used quantitative PCR to measure mRNA levels in the CA1 region of rat hippocampi. The mRNA levels for specific critical proteins were examined, as follows: Protein kinases, PKCγ (0.22), PKCε (0.38), and PKMζ (0.55) mRNAs, and anti-apoptotic, bcl-2 (0.44) and bcl-xl (0.41), were reduced 60 min after hypoxia relative to their expression in tissue not subjected to hypoxia (set to 1.0). Sevoflurane preconditioning prevented the reduction in PKMζ (0.88 vs. 1.0) mRNA levels after hypoxia. Pro-apoptotic BAD mRNA was not significantly changed after hypoxia, even with sevoflurane preconditioning (hypoxia 0.81, sevo hypoxia 0.84 vs. normoxia 1.0). However, BAD mRNA was increased by sevoflurane in non-hypoxic conditions (1.48 vs. 1.0), which may partially explain the deleterious effects of volatile anesthetics under certain conditions. The DNA repair enzyme poly ADP-ribose polymerase 1 (PARP-1) was increased by sevoflurane in tissue not subjected to hypoxia (1.23). PARP-1 mRNA was reduced in untreated tissue after hypoxia (0.21 vs. 1.0); sevoflurane did not improve PARP-1 after hypoxia (0.27). Interestingly, the mRNA level of the cognitive kinase PKMζ, a kinase essential for learning and memory, was the only one protected against hypoxic downregulation by sevoflurane preconditioning. These findings correlate with previous studies that found that sevoflurane-induced improvement of neuronal survival after hypoxia was dependent on PKMζ. Maintaining mRNA levels for critical proteins may provide an important mechanism for preserving neuronal function after stroke.

## 1. Introduction

Sevoflurane is a volatile anesthetic that produces anesthesia quickly and allows rapid recovery from anesthesia. Sevoflurane, unlike other volatile agents, is not pungent and therefore pleasant to breathe, making it an ideal agent for starting anesthesia in children and adults. Hypoxia and ischemia due to stroke, cardiovascular shock, airway obstruction, and other causes lead to neuronal damage and the loss of brain functions [1]. On a cellular level, it causes proteolysis and reduces the amount of key protein kinases, anti-apoptotic, and DNA repair proteins, which may lead to neuronal cell death [2,3,4,5]. Gene expression involves DNA transcription to mRNA, which is then translated into protein. Translational arrest has been shown to contribute to neuronal death after ischemia [2,3]. The transcriptional regulation of genes important for neuronal survival has been less studied. In the current study, we measure the amount of specific mRNA transcripts. Decreased levels of specific mRNAs can be the result of decreased gene expression (transcription) and/or increased degradation. Either way, the result would be a reduction in the amount of protein made from that gene. Importantly, even if the cell’s translational capacity (ability to make protein from mRNA) recovers, the temporal expression of the affected gene remains lost.

Preconditioning induced by a short exposure to a volatile anesthetic, either minutes (immediate preconditioning) or days (delayed preconditioning) before hypoxia or ischemia, has been shown to improve neuronal recovery [5,6,7,8]. Sevoflurane preconditioning may protect neurons by altering the transcriptional downregulation of critical genes or increasing the degradation of specific mRNAs. The current study examines the effect of immediate sevoflurane preconditioning on genes related to intracellular signaling, apoptosis, and DNA repair. These genes have been implicated in survival from hypoxia and ischemia, and their expression following hypoxia may be regulated at the transcriptional level [4,5,6,7,8].

We have previously examined immediate anesthetic preconditioning pathways and found evidence that increased PKMζ protein synthesis was required for improved recovery, and that it was dependent on PKMζ translation and activity [5]. Immediate preconditioning requires mRNA and protein levels to be maintained after the hypoxia or ischemia to support long-term recovery [9]. Other studies found increased levels of certain bcl-2 family anti-apoptotic proteins after ischemia with volatile anesthetic preconditioning [6,10,11].

We hypothesize that hypoxia causes reduced concentrations of proteins critical for neuronal survival by blocking the transcription of their genes. Sevoflurane preconditioning may attenuate the transcriptional downregulation of these genes after hypoxia and ischemia, thereby improving recovery. PKMζ, a kinase important for learning and memory, was protected against downregulation and may contribute to the recovery of cognitive function after hypoxia or ischemia [12]. Maintaining transcription of mRNA for critical proteins may provide an important mechanism for improving recovery after stroke.

## 2. Materials and Methods

### 2.1. Experimental Preparation

The experiments conformed to the NIH “Guide for the Care and Use of Laboratory Animals” and were approved by the Institutional Animal Care and Use Committee of the State University of New York Downstate Health Sciences University (IACUC) (protocol approval number 11-10017). Male 100–120-day-old rats (HSD: Harlan Sprague–Dawley SD) were anesthetized with 2% isoflurane for 2 min in a Plexiglas chamber using an isoflurane vaporizer and then decapitated. Both control and experimental slices were prepared from briefly anesthetized rats as required by the Downstate IACUC. This short period of anesthesia did not appear to have any lasting effects; 15 min of preconditioning with 2% isoflurane did not significantly improve recovery from hypoxia in previous experiments [13]. The hippocampi were removed from the brain, placed in artificial cerebrospinal fluid (aCSF) at 2–4 °C, and sectioned transverse to the long axis of the hippocampus into 400 μm slices using a vibratome [9]. The composition of the aCSF was, in mmol/L NaCl, 126; KCl, 3; KH_2_PO_4_, 1.4; NaHCO_3_, 26; MgSO_4_, 1.3; CaCl_2_, 1.4 glucose, 4; at pH, 7.4. The slices were maintained in oxygenated aCSF (95% O_2_–5% CO_2_) at 25 °C for approximately 2 h before use. The slices were then transferred to a physiological chamber maintained at 37 °C and perfused with aCSF at a rate of 3 mL/min. Both the aCSF and the gas in the chamber were aerated with 95% O_2_–5% CO_2_ [9].

### 2.2. Molecular Biology

Electrophysiological recordings were made from one slice in the chamber and the results of these experiments have been previously reported; the other slices in the chamber were processed for the experiments described in the current paper [5]. The slices were treated with sevoflurane 4% or no sevoflurane for 15 min, followed by a 5 min washout and then 10 min of hypoxia (sevo hypoxia, hypoxia); hypoxia was generated by changing the gas mixture from 95% O_2_–5% CO_2_ to 95%N_2_–5% CO_2_. Sevoflurane was administered to the gas stream with a calibrated vaporizer, the gas was equilibrated with the aCSF and maintained in the atmosphere above the slices in the chamber. Other slices were subjected to the above conditions, except they were not subjected to hypoxia (sevo normoxia, normoxia). The slices were allowed to recover in aCSF, aerated with 95% O_2_ and 5% CO_2_ for 60 min.

#### 2.2.1. Sample Preparation

The CA1 region was microdissected in ice-cold aCSF, homogenized, and preserved in TRIzol reagent. The tissue was kept frozen (−20 °C) until use. Each sample analyzed consisted of the microdissected CA1 regions from hippocampal slices of a single rat on the same grid as that from the electrophysiology experiments (the recorded slice was not used for biochemical analysis).

#### 2.2.2. RNA Isolation and Reverse Transcription

RNA was isolated from the tissue–TRIzol homogenates according to the manufacturer’s instructions. All samples were processed and analyzed within 3 months of freezing, as described below. To prevent genomic DNA contamination, the RNA samples were incubated with RNase-free DNase I (Invitrogen Life Technologies, Carlsbad, CA, USA) and reisolated using the TRIzol protocol. The quality of the RNA isolation was assessed by 1% formaldehyde gel electrophoresis (28S/18S RNA). The quantity and purity of the RNA samples were determined using a spectrophotometer with optical density (OD 260 nm) used for RNA quantification. An OD 260/280 ratio of 1.8–2.0 was used as the cut-off for sample inclusion.

One microgram of RNA from each sample was reverse-transcribed into cDNA using the SuperScript© III First-Strand Synthesis protocol for random hexamer primers (Invitrogen Life Technologies, Carlsbad, CA, USA). Quantitative real-time RT-PCR was performed to measure the mRNA levels; each sample was replicated 3 times and the average of those values was used [14]. To normalize all runs of the RT-PCR to each other, rats of the same age were anesthetized in 2% isoflurane for 2 min, decapitated and their hippocampi removed as described above. However, instead of slicing the hippocampi as above, the whole fresh hippocampus was homogenized and preserved in TRIzol. The RNA was isolated and reverse-transcribed as above. The cDNA was stored at −20 °C and an aliquot was included in each RT-PCR run to standardize the runs to each other.

#### 2.2.3. Quantitative Real-Time RT-PCR

Quantitative real-time RT-PCR (qPCR) was performed using a Stratagene Mx3005P qPCR Machine from Agilent Technologies (Santa Clara, CA, USA). The relative expression of each gene was analyzed using the ΔΔC_T_ method. Amplification reactions consisted of 2.5 ng of cDNA in a 20 μL reaction volume with gene-specific primers (0.5 μM) and BRILLIANT II SYBR^®^ GREEN QPCR MASTER MIX WITH ROX kit (Catalog #600830). Thermocycling conditions were as follows: 50 °C for 2 min (1 cycle), 95 °C for 2 min (1 cycle), followed by 40 cycles of 95 °C for 15 s, 60 °C for 1 min, with a final step of 72 °C for 10 min. At the end of the protocol, a dissociation curve (start temperature of 55 °C) was performed to assess the specificity of amplification. To assess the effect of sevoflurane pretreatment on target gene expression, data were normalized to the housekeeping gene, glyceraldehyde-3-phosphate dehydrogenase (GAPDH). We chose GAPDH as the reference because it is stable under different experimental conditions, including those used in our experiments. In contrast, α-tubulin and β-actin expression might be altered by hypoxia [15,16,17]. In addition, we used GAPDH as a reference protein for Western blots of PKC family proteins in a previous study [5]. The primer sequences used to track the expression of target genes were obtained from the literature and are listed in Table 1 [14,18,19,20].

#### 2.2.4. Statistics

GraphPad Prism 8.0 (GraphPad, San Diego, CA, USA) was used to carry out the statistical analyses; the data were normally distributed. Each value in a group is from 3 to 4 pooled hippocampal CA1 regions of slices from one animal. ANOVA followed by the Student–Neuman–Keuls multiple comparison tests were used to evaluate the data.

## 3. Results

### 3.1. Protein Kinase Genes

Protein kinase C family kinases have been shown to modulate damage from hypoxia and ischemia and are part of the signaling pathway that induces both ischemic and anesthetic preconditioning [21,22,23,24,25]. The mRNA levels of PKCγ (0.22), PKCε (0.38) and PKMζ (0.55), were all significantly reduced 60 min after hypoxia when compared to tissue from normoxic slices (normalized to 1.0) (Figure 1). The PKCγ (0.29), and PKCε (0.44) mRNA levels were also significantly reduced one hour after hypoxia with sevoflurane preconditioning; PKCγ and PKCε did not show significant differences in their mRNA levels when hypoxia after sevoflurane-induced preconditioning was compared to hypoxia alone. However, PKMζ mRNA levels 60 min after hypoxia was significantly greater with sevoflurane-induced preconditioning followed by hypoxia compared to hypoxia alone (0.88 vs. 0.55) (Figure 1). In tissue not subjected to hypoxia, sevoflurane did not significantly alter PKCγ PKCε or PKMζ mRNA levels. Sevoflurane preconditioning improved only PKMζ mRNA expression after hypoxia.

### 3.2. Apoptotic Genes

In neurons, there is a balance of pro-apoptotic proteins and anti-apoptotic proteins, this balance determines whether a neuron will undergo apoptotic cell death [26]. It has been shown that the balance is shifted towards apoptosis by hypoxia and ischemia. We examined whether the levels of mRNAs for pro and anti-apoptotic genes were altered by hypoxia and/or sevoflurane preconditioning. Hypoxia without preconditioning decreased anti-apoptotic bcl-2 (0.44) and bcl-xl (0.41) but did not significantly reduce pro-apoptotic BAD (0.84) mRNA levels compared to tissue not subjected to hypoxia (normoxia, normalized to 1.0). We found that sevoflurane-induced preconditioning did not significantly affect the levels of bcl-2 (0.49 vs. 0.44), bcl-xl (0.58 vs. 0.41) or BAD (0.81 vs. 0.84) mRNA 60 min after hypoxia (Figure 2). Sevoflurane without hypoxia (sevo normoxia) did not significantly alter anti-apoptotic bcl-2 (1.15) or bcl-xl (1.01) mRNA levels compared to untreated normoxic tissue; however, there was a significant increase in the pro-apoptotic BAD mRNA (1.48) with normal oxygen and sevoflurane. The data demonstrate that sevoflurane-induced preconditioning does not significantly alter pro- or anti-apoptotic mRNA levels after hypoxia. However, in the absence of hypoxia, sevoflurane increased pro-apoptotic BAD mRNA; this indicates potential pro-apoptotic effects of volatile anesthetics.

### 3.3. Poly ADP-Ribose Polymerase

We examined mRNA levels for the enzyme poly ADP-ribose Polymerase 1 (PARP-1); this enzyme adds ADP-Ribose to nuclear proteins and facilitates DNA repair [27]. However, PARP1 has also been implicated as contributing to damage during ischemia by increasing ATP utilization, and blockers of its activity have reduced ischemic damage [28,29,30]. Sevoflurane significantly increased PARP1 mRNA levels in normoxic tissue (1.23 vs. 1.0). The mRNA for PARP-1 significantly decreases after hypoxia (0.21); sevoflurane-induced preconditioning did not significantly attenuate this decrease (0.27) (Figure 3). Thus, the reduction in PARP-1 mRNA may play a role in leading to hypoxic damage; however, the mRNA levels of this gene are unlikely to explain the protective effect of sevoflurane-induced preconditioning.

## 4. Discussion

Hypoxia and ischemia due to stroke and other causes frequently result in neuronal damage and loss of brain function. Previous studies, which focused on protein levels and activity, have shown that immediate sevoflurane preconditioning partially protects against neuronal damage and improves cognitive recovery from hypoxia [5,6,7,8,9]. Our previous work demonstrated that the positive effect of sevoflurane preconditioning on recovery from hypoxia and ischemia depended on the translation and activity of PKMζ protein [5,9]. This paper aims to fill a gap in knowledge regarding the effect of hypoxia on gene expression at the transcriptional level, specifically the mRNA levels of critical proteins involved in neuronal function and survival. Additionally, we aspire to shed light on the connection between sevoflurane preconditioning, PKMζ protein expression, and improved neuronal function and recovery following hypoxia.

Hypoxia and ischemia induce proteolysis and free radical damage to nucleic acids, critical proteins, and lipids, leading to neuronal damage and death by either necrosis or apoptosis [1,4]. Neurons that receive a moderate hypoxic/ischemic insult, such as those in the penumbra of a stroke, primarily undergo apoptosis, a delayed cell-death pathway that can be reversed after it is initiated. These neurons may recover from such damage if gene expression replaces the damaged proteins. Certain treatments, such as sevoflurane preconditioning, enhance recovery from hypoxia/ischemia [1,9]. Recent data indicate that ribosomes are inactivated following hypoxia or ischemia, leading to an inhibition of translation and cell death via apoptosis [2,3].

This study examined the effect of hypoxia on gene expression by measuring the concentration of specific mRNA transcripts. DNA transcription and mRNA processing occur in the nucleus upstream of translation, which occurs in the cytoplasm. A reduction in mRNA levels could be the result of repressed transcription in the nucleus, increased mRNA degradation in the cytoplasm, or both. The relevance of this to neuronal recovery is that, in either case, restoring translation would not by itself overcome a transcriptional block of gene expression for protein kinase and anti-apoptotic genes that are important for neuronal survival.

Here we report that hypoxia significantly reduced the mRNA for proteins required for neuronal function and survival. For example, hypoxia led to a decrease in the mRNA levels of the anti-apoptotic genes bcl-2 and bcl-xl. With no significant change in the pro-apoptotic gene BAD, this balance would shift towards apoptosis after hypoxia. We examined mRNA expression 1 h after hypoxia; it is unclear how long anti-apoptotic genes remain downregulated after this, or whether sevoflurane preconditioning speeds up the recovery of their transcription. These possibilities require further study in a different model because adult brain slices are not useful for long-term studies.

In tissue not subjected to hypoxia, sevoflurane increased pro-apoptotic BAD mRNA levels, which might explain some of the deleterious effects found with sevoflurane anesthesia under certain conditions [31,32,33]. This selective downregulation of mRNAs from anti-apoptotic genes indicates that restoring translation shortly after hypoxia is unlikely to be sufficient to prevent apoptotic cell death and that selective downregulation may play an important role in post-hypoxic apoptotic cell death. Sevoflurane, along with enhanced PKMζ expression, is likely acting to block pathways downstream of these apoptotic genes. Interestingly, of the many critical genes examined, sevoflurane preconditioning rescued the mRNA expression of just one: PKMζ, an atypical protein kinase C required for long-term potentiation and the maintenance of memory [34,35].

There are two types of preconditioning: immediate preconditioning is applied minutes before a hypoxic/ischemic event and delayed preconditioning is applied a day in advance. Our study examines immediate preconditioning. However, both types of preconditioning need to have effects both immediately after the hypoxia and for days later; the later requires maintained gene transcription. Our slice study demonstrated short-term protection, but we have also demonstrated prolonged protection after immediate sevoflurane preconditioning [5,9]. Preserved gene transcription is required for long-term protection.

A unique aspect of our study is that we measured mRNA levels selectively in the hippocampal CA1 region, a region particularly sensitive to hypoxic and ischemic damage. Importantly, this is the brain region we previously examined for physiological and histological recovery [5,9]. The mRNA was measured one hour after hypoxia; this is a time in which the tissue is highly vulnerable and likely to encounter genetic damage from free radicals generated upon reoxygenation [1].

We examined the relative amounts of mRNA from three classes of proteins: kinases of the PKC family, pro- and anti-apoptotic agents, and a DNA repair enzyme using quantitative PCR. The choice of genes was based on previous studies that found altered protein concentrations of these genes after hypoxia or ischemia with anesthetic preconditioning [5,6,10,11,15,36,37].

An in vivo study of immediate sevoflurane preconditioning found prolonged protection 6 weeks after 10 min of global cerebral ischemia [9]. Using hippocampal slice preparations, we found that sevoflurane preconditioning improved the recovery of the resting potential 60 min after hypoxia in CA1 pyramidal cells pretreated with sevoflurane compared to untreated tissue subjected to hypoxia. PKMζ protein was neither reduced nor increased 1 h following hypoxia treatment alone. However, the PKMζ protein concentration was increased in tissue protected by sevoflurane preconditioning [5]. Tissue from these experiments was saved and the mRNA levels of selected genes were examined in the current study.

In slices not treated with sevoflurane, PKMζ mRNA, but not PKMζ protein concentrations, were reduced 1 h after hypoxia. This could be related to the longer half-life of PKMζ protein compared to the other PKC family enzymes [12]. The increased PKMζ protein concentration following sevoflurane and hypoxia compared with hypoxia alone would require PKMζ mRNA levels to have been maintained.

The rescue of PKMζ mRNA levels in preconditioned samples documented in the current study explains the elevated PKMζ protein levels measured in the previous study and likely contributed to the improved neuronal survival observed with sevoflurane preconditioning. Interestingly, mTOR pathway activation is important for sevoflurane-induced protection and other downstream effects of this pathway and, in addition to PKMζ, may play an important protective role [5,38]. Zeta inhibitory peptide (ZIP), an inhibitor of PKMζ activity; rapamycin, an mTOR pathway inhibitor; and cycloheximide, a protein synthesis inhibitor, each blocked sevoflurane protection and reduced the concentration of PKMζ protein in the sevoflurane preconditioning group [5]. This finding is supported by a recent study that showed that sevoflurane protection of heart tissue from ischemic injury also requires activation of the mTOR pathway [38].

In a study by Libien et al., reduced PKCγ and PKCε protein concentrations were measured 1 h after 10 min of hypoxia; however, the mechanism for the reduced expression of these kinases was unclear [39]. The current study indicates the reduced PKCγ and PKCε protein is due, at least in part, to a significant depression of PKCγ and PKCε mRNA one hour after hypoxia; this supports the importance of maintaining gene expression directly after hypoxia.

The enzyme PARP-1 adds ADP-Ribose to nuclear proteins and facilitates the repair of damaged DNA, as well as contributing to activity-dependent gene expression required for synaptic plasticity [27,40]. In the absence of hypoxia, sevoflurane increased PARP1 mRNA levels; this may lead to enhanced DNA repair or increased synaptic plasticity during normoxia. Hypoxia reduces PARP1 mRNA, which may lead to decreased DNA repair under circumstances in which DNA damage is intensified. The increase in damaged DNA would contribute to increased apoptosis. Sevoflurane preconditioning did not prevent the hypoxia-induced reduction in PARP1 mRNA and therefore cannot explain recovery after hypoxia.

The effects of sevoflurane-induced preconditioning on gene expression may indicate important long-term regulatory mechanisms that lead to protection hours, days, and weeks later. Preventing the fall in PKMζ mRNA after hypoxia with sevoflurane pre-conditioning may allow increased PKMζ protein expression, leading to the prolonged protection against neuronal death that we found in a previous study [8]. Given the importance of PKMζ for long-term plasticity and memory formation, the preservation of PKMζ mRNA may be a promising approach to improving cognitive function following stroke [11].

## 5. Conclusions

In conclusion, sevoflurane increased the mRNA levels of BAD in tissue not subjected to hypoxia, which may explain some of the deleterious effects found after sevoflurane anesthesia. Protein kinase, anti-apoptotic, and DNA repair mRNA levels are downregulated after hypoxia, and this may contribute to neuronal cell death. Significantly, the volatile anesthetic sevoflurane rescues PKMζ mRNA levels in the post-hypoxic period, which correlates with the improved neuronal survival found in earlier studies. We conclude that the reduction in mRNAs for critical proteins may be an important determinant of neuronal dysfunction after hypoxia. Improving the expression of critical proteins by maintaining mRNA levels may provide a novel mechanism for improving recovery from hypoxic/ischemic neuronal damage due to stroke.

## Figures and Tables

**Figure 1 neurosci-06-00009-f001:**
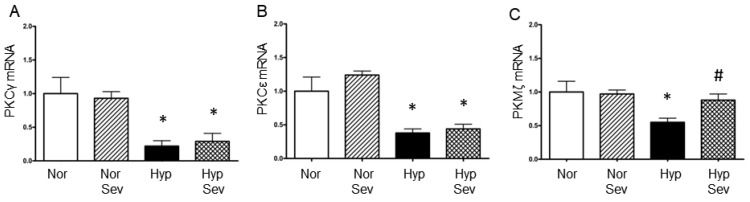
The effect of sevoflurane-induced preconditioning on mRNA levels of Protein Kinase C family genes after hypoxia. (**A**,**B**) There was a significant reduction of PKCγ and PKCε mRNA after hypoxia with and without sevoflurane when compared to normal oxygen. (**C**) There was a significant reduction of PKMζ mRNA after hypoxia without sevoflurane when compared to normal oxygen; sevoflurane significantly increased PKMζ mRNA after hypoxia when compared to hypoxia without sevoflurane. Hippocampal slices were perfused with normoxic aCSF (95% O_2_/5%CO_2_) and then subjected to either 0% sevoflurane (Nor) or 4% sevoflurane (Sev) for 15 min, washed for 5 min and then subjected to 10 min of hypoxia (95% N_2_/5% CO_2_) (Hyp or Hyp Sev) or normal oxygen (Nor or Nor Sev) followed by 60 of recovery in normoxic aCSF. The CA1 region of the hippocampus was dissected and analyzed using quantitative RT PCR. Data were analyzed with an ANOVA followed by the Student–Neuman–Keuls multiple comparison test; * *p* < 0.05 vs. normoxia group; # *p* < 0.01 vs. hypoxia group. Values are the mean ± the standard error of the mean (normoxia *n* = 4; sevo normoxia *n* = 5; hypoxia *n* = 7; sevo hypoxia *n* = 7).

**Figure 2 neurosci-06-00009-f002:**
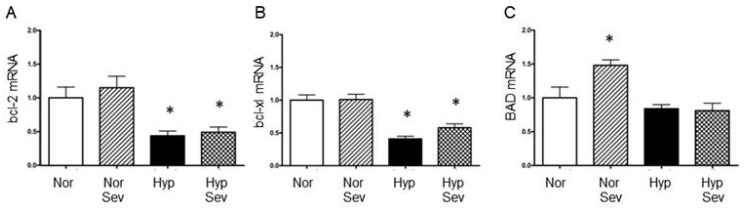
The effect of sevoflurane-induced preconditioning on mRNA levels of bcl-2 family pro- and anti-apoptotic genes. (**A**,**B**) There was a significant reduction in anti-apoptotic bcl-2 and bcl-xl mRNA after hypoxia with and without sevoflurane when compared to normal oxygen. (**C**) There was a significant increase in pro-apoptotic BAD mRNA in normal oxygen with sevoflurane when compared to normal oxygen without sevoflurane. Hippocampal slices were perfused with normoxic aCSF (95% O_2_/5%CO_2_) and then subjected to either 0% sevoflurane (Nor) or 4% sevoflurane (Sev) for 15 min, washed for 5 min and then subjected to 10 min of hypoxia (95% N_2_/5% CO_2_) (Hyp or Hyp Sev) or normal oxygen (Nor or Nor Sev), followed by 60 of recovery in normoxic aCSF. The CA1 region of the hippocampus was dissected and analyzed using quantitative RT PCR. Data were analyzed with an ANOVA followed by the Student–Neuman–Keuls multiple comparison test; * *p* < 0.01 vs. normoxia group. Values are the mean ± the standard error of the mean (normoxia *n* = 4; sevo normoxia *n* = 5; hypoxia *n* = 7; sevo hypoxia *n* = 7).

**Figure 3 neurosci-06-00009-f003:**
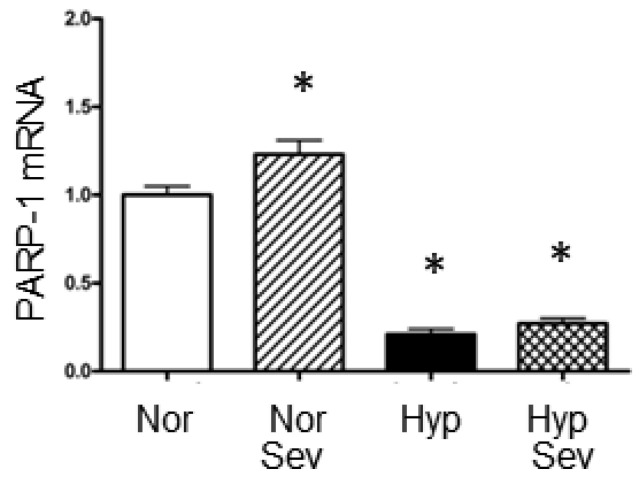
The effect of sevoflurane-induced preconditioning on mRNA levels of poly ADP-ribose polymerase 1 (PARP1). There was a significant increase in PARP-1 mRNA with sevoflurane in normal oxygen compared to normal oxygen without sevoflurane; there was a significant decrease in PARP-1 mRNA after hypoxia with or without sevoflurane. Hippocampal slices were perfused with normoxic aCSF (95% O_2_/5%CO_2_) and then subjected to either 0% sevoflurane (Nor) or 4% sevoflurane (Sev) for 15 min, washed for 5 min and then subjected to 10 min of hypoxia (95% N_2_/5% CO_2_) (Hyp or Hyp Sev) or normal oxygen (Nor or Nor Sev) followed by 60 of recovery in normoxic aCSF. The CA1 region of the hippocampus was dissected and analyzed using quantitative RT PCR. Data were analyzed with an ANOVA followed by the Student–Neuman–Keuls multiple comparison test; * *p* < 0.01 vs. normoxia group. Values are the mean ± the standard error of the mean (normoxia *n* = 4; sevo normoxia *n* = 5; hypoxia *n* = 7; sevo hypoxia *n* = 7).

**Table 1 neurosci-06-00009-t001:** Sequences of primers for quantitative real-time RT-PCR and size of products.

mRNA	Forward Primer Sequence (5′→3′)	Reverse Primer Sequence (5′→3′)	Size of Product (Bp)
Bad	GCT TAG CCC TTT TCG AGG AC	GAT CCC ACC AGG ACT GGA T	200
Bcl-xL	GGT GAG TCG GAT TGC AAG TT	GAG CCC AGC AGA ACT ACA CC	198
Bcl2	AGGGGCTACGAGTGGGATAC	TCAGGCTGGAAGGAGAAGATG	86
PKMζ	GGC TCC TTA AAG GGA CGG AA	TGC TCT ACC GAA GGT GGG C	54
PKCε	CCC CTT GTG ACC AGG AAC TA	GCC TTT GCC TAA CAC CTT GA	203
PKCγ	TTC TTC AAG CAG CCA ACC TT	TGT AGC TGT GCA GAC GGA AC	202
Parp 1	AGTATGCCAAGTCCAACAGGAGCA	ATCATACCCAGTTGCGGCTTCTCT	114
GAPDH	GAACATCATCCCTGCATCCA	CCAGTGAGCTTCCCGTTCA	70

## Data Availability

The datasets used and/or analyzed during the current study are available from the corresponding author on reasonable request.

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
