# Peer review of "Sevoflurane Preconditioning Rescues PKMζ Gene Expression from Broad Hypoxia-Induced mRNA Downregulation Correlating with Improved Neuronal Recovery"

_neurosci, 2025, doi:10.3390/neurosci6010009_

Round 1
Reviewer 1 Report
Comments and Suggestions for Authors
The manuscript (neurosci-3350428) examined the effects of sevoflurane preconditioning on gene expression of a set of protein kinases (PKCγ, PKCε, and PKMζ), apoptosis-related proteins (bcl-2, bcl-xL and BAD) and poly ADP-ribose polymerase 1 (PARP-1) in an in-vitro model of hypoxia-exposed rat hippocampal slices. The main finding is that sevoflurane preconditioning reversed the downregulation of PKMζ expression after hypoxia. It was concluded that improving the expression of critical proteins by maintaining mRNA transcription may provide a novel mechanism for improving recovery from hypoxic/ischemic neuronal damage due to stroke.
Overall, while the topic of the study on the neuroprotective effects of sevoflurane preconditioning is interesting and the manuscript is generally well written, my main problem with the manuscript is that, according to the statement on line 274-276 that “Tissue from these experiments was saved and the mRNA expression of selected genes was examined in the current study”, it appears that the current experiments were performed on tissues collected at least earlier than 2012 when works on the brain slices were published (ref#4: Wang et al. 2012). Therefore, when was the current study carried out? If the experiments of RNA extraction and qPCR were performed recently on frozen tissue collected more than ten years ago, I would have problem to trust any of the results.
I also have a couple of more comments for the authors:
1. The authors appear to suggest that decreased mRNA levels are due to transcriptional downregulation. Is it possible that increased mRNA turnover could be at least partly involved?
2. The Methods section should describe the statistics employed.
Author Response
Thank you for your comment we have tried to address them as follows:
”, it appears that the current experiments were performed on tissues collected at least earlier than 2012 when works on the brain slices were published (ref#4: Wang et al. 2012). Therefore, when was the current study carried out? If the experiments of RNA extraction and qPCR were performed recently on frozen tissue collected more than ten years ago, I would have problem to trust any of the results.
This is an important point and we now state clearly the samples were processed and analyzed within 3 months of freezing. Ln 124
So why the delay in publishing, this is the long story. We initially tried to publish in 2014 but at that time the scientist with molecular experience, who caried out the study (j. Hou) had left the lab for clinical residency and I wrote up the study. At that time my experience in molecular biology was limited and the manuscript was lacking. My lab moved away from hypoxia and ischemia studies due to funding issues and began looking at the effects of neonatal anesthesia on behavior and molecular biology with a different scientist (D Lin), the eliminated self cites were her work. The current paper was put aside and I only began looking at it during covid when my lab emptied out. I received a solicitation for papers from this journal and revised the paper. I have gotten more knowledgeable about molecular biology and I think this manuscript’s writing is greatly improved. We always thought the results were important and worth publishing.
- The authors appear to suggest that decreased mRNA levels are due to transcriptional downregulation. Is it possible that increased mRNA turnover could be at least partly involved?
We agree and now state “due to decreased transcription and/or increased degradation of mRNA (ln 58; ln 267. We also changed mRNA expression to mRNA levels throughout the paper.
- The Methods section should describe the statistics employed.
Thank you this section must have been eliminated accidently, I have added it back ln162
2.2.4. Statistics
GraphPad Prism 8.0 (GraphPad, San Diego, CA) was used to carry out the statistical analyses; the data were normally distributed. Each value in a group is from 3-4 pooled hippocampal CA1 regions of slices from one animal. An ANOVA followed by the Student-Neuman-Keuls multiple comparison tests were used to evaluate the data.
Reviewer 2 Report
Comments and Suggestions for Authors
Dear Authors,
I have read your submitted manuscript entitled "Sevoflurane preconditioning rescues PKMz gene expression from broad hypoxia-induced transcriptional downregulation correlating with improved recovery" and I would like to make some comments for its improvement:
-The Abstract is quite long whereas the Introduction section needs to be expanded including information on the Sevoflurane and its use. In addition, the later should be better structured, with the aims and the objectives of the current study clearly presented at the end.
-In the Discussion section, there should be a clearer presentation and explanation of the differences in the effect of the drug on the various genes; as it is now, it is quite difficult for the reader to understand.
Thank you for considering my recommendations. I wish you good luck with the publication of your research work.
Kind regards,
The Reviewer
Author Response
Reviewer 2
Thank you for your comment we have tried to address them as follows:
-The Abstract is quite long whereas the Introduction section needs to be expanded including information on the Sevoflurane and its use. In addition, the later should be better structured, with the aims and the objectives of the current study clearly presented at the end.
We edited the abstract We highlight the additions but did not highlight the deletions, rewording or rearrangements in the abstract.
We have added information about sevoflurane to the introduction:
Sevoflurane is a volatile anesthetic that produces anesthesia quickly and also allows rapid recovery after anesthesia. Sevoflurane, unlike other volatile agents, is not pungent and therefore pleasant to breath; this makes it an ideal agent for starting anesthesia in children and adults. Ln 46-49. We present our hypothesis and aims of the study in the last paragraph of the introduction. Ln 77-84
In the Discussion section, there should be a clearer presentation and explanation of the differences in the effect of the drug on the various genes; as it is now, it is quite difficult for the reader to understand.
We have added a section to the beginning of the discussion to address this. Ln 243
Hypoxia and ischemia due to stroke and other causes frequently result in neuronal damage and loss of brain function. Previous studies which focused on protein levels and activity, have shown that immediate sevoflurane preconditioning partially protects against neuronal damage and improves cognitive recovery from hypoxia[5-9]. Our previous work demonstrated that the positive effect of sevoflurane preconditioning on recovery from hypoxia and ischemia depended on the translation of and activity of PKMζ protein[5, 9]. This paper aims to fill a gap in knowledge regarding the effect of hypoxia on gene expression at the transcriptional level--specifically, the mRNA levels of critical proteins involved in neuronal function and survival. Additionally, we aspire to shed light on the connection between sevoflurane preconditioning, PKMζ protein expression and improved neuronal function and recovery following hypoxia.
We have reworked the discussion and eliminated repetition. Major additions are highlighted but deletions and rearrangements are not highlighted. We think this clarifies the discussion.
Round 2
Reviewer 1 Report
Comments and Suggestions for Authors
I am satisfied with the response and have no more questions.